# Hyperbolic Random Forests

**Lars Doorenbos**                                                                *lars.doorenbos@unibe.ch*
*University of Bern*

**Pablo Márquez-Neila**                                                          *pablo.marquez@unibe.ch*
*University of Bern*

**Raphael Sznitman**                                                          *raphael.sznitman@unibe.ch*
*University of Bern*

**Pascal Mettes**                                                                  *p.s.m.mettes@uva.nl*
*University of Amsterdam*

**Reviewed on OpenReview:** *https://openreview.net/forum?id=pjKcIzvXWR*

## Abstract

Hyperbolic space is becoming a popular choice for representing data due to the hierarchical structure — whether implicit or explicit — of many real-world datasets. Along with it comes a need for algorithms capable of solving fundamental tasks, such as classification, in hyperbolic space. Recently, multiple papers have investigated hyperbolic alternatives to hyperplane-based classifiers, such as logistic regression and SVMs. While effective, these approaches struggle with more complex hierarchical data. We, therefore, propose to generalize the well-known random forests to hyperbolic space. We do this by redefining the notion of a split using horospheres. Since finding the globally optimal split is computationally intractable, we find candidate horospheres through a large-margin classifier. To make hyperbolic random forests work on multi-class data and imbalanced experiments, we furthermore outline new methods for combining classes based on the lowest common ancestor and class-balanced large-margin losses. Experiments on standard and new benchmarks show that our approach outperforms both conventional random forest algorithms and recent hyperbolic classifiers.

## 1 Introduction

Machine learning in hyperbolic space is gaining more and more attention, and hyperbolic representations of data have already found success in numerous domains, such as natural language processing (Nickel & Kiela, 2017; Tai et al., 2022) computer vision (Ahmad & Lecue, 2022; Khrulkov et al., 2020; Ghadimi Atigh et al., 2021), graphs (Chami et al., 2019; Liu et al., 2019; Sun et al., 2021b), recommender systems (Sun et al., 2021a), and more. Hyperbolic space is a natural choice for data with a hierarchical structure due to the fact that the available space grows exponentially when moving away from the origin. Therefore, it can be seen as a continuous version of a graph-theoretical tree (Nickel & Kiela, 2018).

With the rise of datasets embedded in hyperbolic space comes a need for algorithms that can successfully operate on them (Cho et al., 2019). As a result, many methods specifically designed for hyperbolic space have been proposed that tackle a variety of machine learning tasks, such as clustering (Monath et al., 2019), regression (Marconi et al., 2020), and classification (Ganea et al., 2018b; Chien et al., 2021; Cho et al., 2019; Pan et al., 2023; Weber et al., 2020).

Current hyperbolic classification algorithms, such as hyperbolic support vector machines (Fan et al., 2023) or hyperbolic logistic regression (Ganea et al., 2018b), have shown promising results, but still struggle with complex datasets. In Euclidean space, there is a long history of success of tree-based random forest algorithms

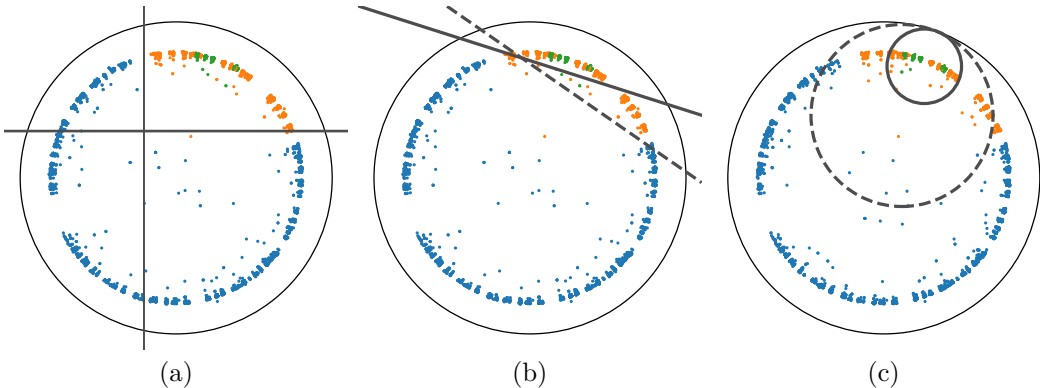

Figure 1: **Motivation for hyperbolic random forests.** Two splits of (a) an axis-aligned Euclidean, (b) an oblique Euclidean, and (b) a hyperbolic decision tree of depth two. The data is a continuous embedding of a tree split into three nested classes. The inductive biases of linear decision boundaries are inappropriate for efficiently capturing the underlying geometry, and more splits are needed to be effective.

in this regard, and they remain among the most popular learning algorithms for many data types (Caruana & Niculescu-Mizil, 2006; Fernández-Delgado et al., 2014). We argue that decision trees and, by extension, random forests are well-suited for hyperbolic space due to their shared hierarchical structure. Figure 1 illustrates the need for a classifier that fits the underlying geometry and the benefits of a hyperbolic tree-based classifier. Euclidean hyperplane splits (Figure 1(a/b)) are ineffective at capturing the structure of the data. In contrast, our hyperbolic splits (Figure 1(c)) are more appropriate, where the nested splits clearly show how the smallest green class is a subclass of the larger orange one.

Tree-based algorithms are built by recursively applying a splitting function; hence, to generalize the concept of a tree-based classifier to hyperbolic space, we need to define a hyperbolic splitting function. For this purpose, we propose to use *horospheres*, which share several desirable properties with the hyperplanes used in Euclidean trees. Due to a combinatorial explosion, finding the optimal horosphere by enumeration is computationally intractable. We propose to tackle data splitting as a binary horosphere classification task. To deal with multiple classes in a hierarchically consistent manner, we extend our splitting operator using hyperclasses. We show how to obtain hyperclasses through clustering in hyperbolic space with the lowest common ancestor. We also outline a class-balancing large-margin loss for dealing with long-tailed data.

We extensively evaluate our approach on canonical benchmarks from previous works on hyperbolic classification. Additionally, we introduce three new multi-class classification experiments. We show that our method is superior to both competing hyperbolic classifiers as well as their Euclidean counterparts in these settings. Summarized, our contributions are (1) a generalization of random forests to hyperbolic space using horospheres, dubbed HoroRF, (2) two extensions to enable effective learning in multi-class and imbalanced settings, and (3) a thorough evaluation of HoroRF, showing its advantage over other methods, both Euclidean and hyperbolic.

## 2 Related Work

### 2.1 Hyperbolic machine learning

Machine learning in hyperbolic space has gained traction due to its inherent hierarchical and compact nature. Foundational work showed that hyperbolic space is superior to Euclidean space for embedding hierarchies in a continuous space, allowing for embeddings with minimal distortion (Nickel & Kiela, 2017; Ganea et al., 2018a). Empowered by these results, learning with hyperbolic embeddings has recently been successfully used for various problems. Chami et al. (2019) and Liu et al. (2019) showed how to generalize graph networks to hyperbolic space, while Tifrea et al. (2019) showed the potential of hyperbolic word embeddings. In the visual domain, hyperbolic embeddings have been shown to improve image segmentation (Ghadimi Atigh

et al., 2022), zero-shot recognition (Liu et al., 2020), image-text representation learning (Desai et al., 2023), and more. Hyperbolic embeddings have also been effective for biology (Klimovskaia et al., 2020) and in recommender systems (Sun et al., 2021a). We refer to recent surveys for a more complete overview (Peng et al., 2021; Yang et al., 2022; Mettes et al., 2023).

For classification specifically, various traditional classifiers such as logistic regression (Ganea et al., 2018b), neural networks (Ganea et al., 2018b; Shimizu et al., 2021; van Spengler et al., 2023), and support vector machines (SVM) (Cho et al., 2019; Fan et al., 2023; Fang et al., 2021; Weber et al., 2020) now have counterparts in hyperbolic space. In this work, we strive to go beyond single hyperplane/gyroplane decision boundaries per class and bring random forests to hyperbolic space.

Our method is built on the concept of horospheres, which are the level sets of the Busemann function to ideal points. Ideal points have recently succeeded in multiple areas, such as supervised classification (Ghadimi Atigh et al., 2021) and self-supervised learning (Durrant & Leontidis, 2023). Horospheres, in particular, have found applications in dimensionality reduction (Chami et al., 2021), as well as generalizing SVMs (Fan et al., 2023) and neural networks (Sonoda et al., 2022) to hyperbolic space, but we are the first to use them as building blocks for random forests. Concurrent with our work, Chlenski et al. (2023) also propose a hyperbolic random forest model where they rely on geodesics rather than horospheres and work in the hyperboloid model of hyperbolic space instead of the Poincaré disk model.

## 2.2 Random Forests

Random Forests remain a popular choice of classifier to this day due to their high performance, speed, and insensitivity to hyperparameters (Grinsztajn et al., 2022). In their original paper (Breiman, 2001), two versions of random forests were proposed: one with axis-aligned splits based on a single feature and one with oblique splits based on linear combinations of features. While the optimal axis-aligned split can be found by exhaustive search, finding the optimal oblique split is NP-complete (Heath et al., 1993). As a result, many works have developed heuristics to find good oblique splits in a reasonable time.

One line of work makes use of meta-heuristics such as hill climbing (Murthy et al., 1993), simulated annealing (Heath et al., 1993), or genetic algorithms (Cantú-Paz & Kamath, 2003). Other approaches train one or more binary classifiers at every node, choosing among the resulting hyperplanes to split the data. Examples include using linear discriminant analysis, ridge regression (Menze et al., 2011), or support vector machines (Do et al., 2010). For multi-class cases, typically, heuristics are employed to partition them into two hyperclasses (Katuwal & Suganthan, 2018). We follow the binary classifier approach for our hyperbolic random forests and introduce a hyperclass heuristic specifically designed for hyperbolic space based on hierarchical clustering Chami et al. (2020).

# 3 Hyperbolic Random Forests

## 3.1 The Poincaré ball model

We follow the convention from previous works using horospheres (Chami et al., 2021; Fan et al., 2023) and hyperbolic machine learning more broadly (Ganea et al., 2018b; Khrulkov et al., 2020; Chien et al., 2021; Shimizu et al., 2021) and make use of the Poincaré ball model of hyperbolic space. The Poincaré ball model is defined by the metric space $(\mathbb{B}_c^n, g_c^{\mathbb{B}})$ for a given negative curvature $c$, which we set to 1 throughout this work, where

$$\mathbb{B}_1^n = \{\mathbf{x} \in \mathbb{R}^n : \|\mathbf{x}\| < 1\} \tag{1}$$

with $\|\cdot\|$ the standard Euclidean norm, and

$$g_1^{\mathbb{B}}(\mathbf{x}) = \frac{2}{1 - \|\mathbf{x}\|^2} \mathbf{I}_n. \tag{2}$$

From here on, we will omit the curvature subscript for clarity. The distance between two points is given by

$$d_{\mathbb{B}}(\mathbf{a}, \mathbf{b}) = \operatorname{arcosh}\left(1 + 2\frac{\|\mathbf{a} - \mathbf{b}\|^2}{(1 - \|\mathbf{a}\|^2)(1 - \|\mathbf{b}\|^2)}\right), \tag{3}$$

which is the length of the geodesic arc connecting them. Extending geodesics to infinity in one direction leads to a point on the boundary of the Poincaré ball. These points on the boundary are known as *ideal points*. The set of all ideal points thus lies on the hypersphere $\mathbb{S}^{n-1}$, and they can be interpreted as directions in hyperbolic space (Chami et al., 2021).

### 3.2 HoroRF

To generalize random forests to hyperbolic space, we require a way to split data points recursively for any number of classes, regardless of the class frequency distribution, into two partitions with low impurity. In Euclidean space, this splitting function outputs hyperplanes to split the data. For hyperbolic random forests, we propose to use horospheres.

#### 3.2.1 Formalization

Horospheres are the level sets of the Busemann function (Busemann, 1955), which calculates the normalized distance to infinity in a given direction. It can be expressed in closed form in the Poincaré model:

$$B_{\mathbf{w}}(\mathbf{x}) = \log \frac{\|\mathbf{w} - \mathbf{x}\|^2}{1 - \|\mathbf{x}\|^2}. \tag{4}$$

As a result, horospheres $\pi_{\mathbf{w},b}$ are parameterized by an ideal point $\mathbf{w} \in \mathbb{S}^{n-1}$, and a distance to that point $b \in \mathbb{R}$, which is the radius of the horosphere. Our goal is to find the optimal horosphere $\pi'_{\mathbf{w},b}$ that minimizes the impurity of the resulting partitions or, equivalently, maximizes the information gain. Consider a node $S$ with data $\{(\mathbf{x}_i, y_i)\}_{i=1}^N$ with $\mathbf{x} \in \mathbb{B}^n$ and $y \in Y$. The optimal horosphere is given by

$$\pi'_{\mathbf{w},b} = \arg\max_{\pi_{\mathbf{w},b}} (H(S) - I_{in} - I_{out}), \tag{5}$$

where $H$ is an impurity measure, and $I_{in}$ and $I_{out}$ give the impurity of the set of points inside and outside of the horosphere, respectively:

$$I_{in} = \frac{N_{in}}{N} H(\{\mathbf{x} \in S | B_{\mathbf{w}}(\mathbf{x}) < b\}),$$
$$I_{out} = \frac{N_{out}}{N} H(\{\mathbf{x} \in S | B_{\mathbf{w}}(\mathbf{x}) \geq b\}). \tag{6}$$

Here, $N_{in}$ and $N_{out}$ denote the number of samples inside and outside the horosphere. Similar to oblique decision trees in Euclidean space, finding the globally optimal horosphere is computationally infeasible. As such, we need to find approximate solutions.

#### 3.2.2 Finding approximate solutions

As is conventional in (oblique) decision tree literature (*e.g.*, Do et al. (2010); Menze et al. (2011); Katuwal et al. (2020)), we employ a binary classifier at every node to find candidate solutions. We need a hyperbolic classifier that will allow us to find splits that fit the underlying geometry. Large margin classifiers based on horospheres guarantee a globally optimal solution, bypassing hyperbolic hyperplane-based methods that fail to converge (Weber et al., 2020) or are limited to two dimensions (Chien et al., 2021). Moreover, the state-of-the-art hyperbolic SVM, HoroSVM, is based on horospheres and outperforms hyperbolic hyperplane-based methods (Fan et al., 2023). For this reason, we build upon HoroSVM. For a labeled dataset $D = \{(\mathbf{x}_i, y_i)\}_{i=1}^N$ with $\mathbf{x}_i \in \mathbb{B}^n$ and $y_i \in \{-1, 1\}$, it optimizes the convex loss function

$$\ell(\mu, \mathbf{w}, o; D) = \frac{1}{2}\mu^2 + C \sum_{i=1}^N \max\left(0, 1 - y_i(\mu B_{\mathbf{w}}^{-1}(\mathbf{x}_i) - o)\right), \tag{7}$$

where

$$B_{\mathbf{w}}^{-1}(\mathbf{x}) = \log \frac{1 - \|\mathbf{x}\|^2}{\|\mathbf{w} - \mathbf{x}\|^2}, \tag{8}$$

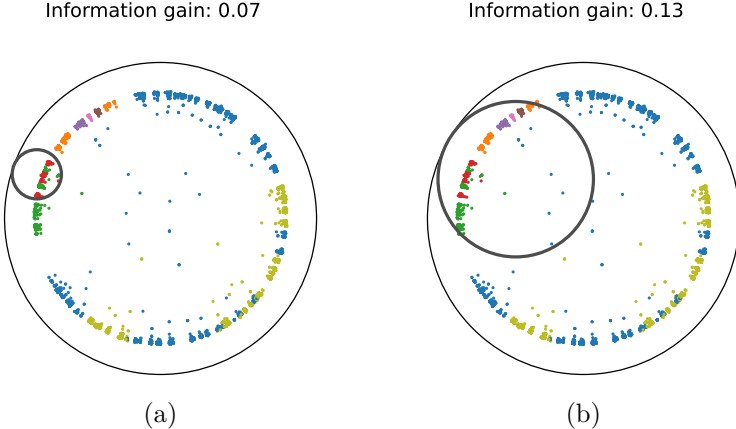

Figure 2: **Information gain from hierarchical splits.** The best (a) one-versus-rest split and (b) split found with our hyperclass heuristic. We find splits with a higher information gain by considering splits where more than one class is seen as positive.

with $\mu, o \in \mathbb{R}^+$ and $\mathbf{w} \in \mathbb{S}^{n-1}$. The slack hyperparameter $C$ controls the trade-off between misclassification tolerance and margin size. Solving for HoroSVM thus results in the three parameters $\mu, \mathbf{w}$, and $o$. In order to transform these results into a horosphere $\pi_{\mathbf{w},b}$, we set $b = -\frac{o}{\mu}$ and use $\mathbf{w}$ directly.

We repeat the classification $K$ times in a one-versus-rest setting, with $K$ the number of classes, and compute the information gain for all resulting horospheres. We select the horosphere with the highest information gain to split the data. We refer to this splitting procedure as the `HoroSplitter`. While this base version of the `HoroSplitter` already achieves competitive results, it naturally has limited capacity to deal with imbalanced data, and the one-versus-rest set-up limits its effectiveness in multi-class settings. Therefore, we introduce two additional components to the `HoroSplitter` that make it more appropriate specifically as a building block for hyperbolic decision trees.

### 3.2.3  Combining classes

The default `HoroSplitter` uses a one-versus-rest approach to deal with multi-class data. As a result, it can only find one-versus-rest splits. Nonetheless, a horosphere split with a high information gain could have more than one class in either partition. To find good splits with multiple classes while avoiding searching over all possible combinations, only the most promising combinations should be evaluated. For this, we design a heuristic that transforms the multi-class problem into a hierarchical set of binary problems by iteratively grouping classes into two hyperclasses based on their lowest common ancestor (LCA). Then, these binary combinations are added to the pool of one-versus-rest experiments and evaluated for their information gain.

Recall the connection of hyperbolic space to graph-theoretical trees. In a tree, similar leaf nodes lie in a small subtree and have their LCA at a high depth. In contrast, dissimilar nodes will have their LCA close to the root of the tree. We exploit the analog of this property in hyperbolic space to cluster together the most similar classes first, as similar classes are likely to be capturable by a single horosphere.

We represent each cluster by its hyperbolic mean. As computing the average in the Poincaré model involves the computationally expensive Fréchet mean (Lou et al., 2020), we use the Einstein midpoint after transforming the data to Klein coordinates instead. Specifically, data is transformed into Klein space $\mathbb{K}$ with

$$\mathbf{x}_{\mathbb{K}} = \frac{2\mathbf{x}_{\mathbb{B}}}{1 + \|\mathbf{x}_{\mathbb{B}}\|^2}, \tag{9}$$

and we compute the average $\phi_y$ for class $y$ using

$$\phi_y = \frac{\sum_{i=1}^N \mathbb{1}_{[y_i=y]} \gamma_i \mathbf{x}_i}{\sum_{i=1}^N \gamma_i}, \tag{10}$$

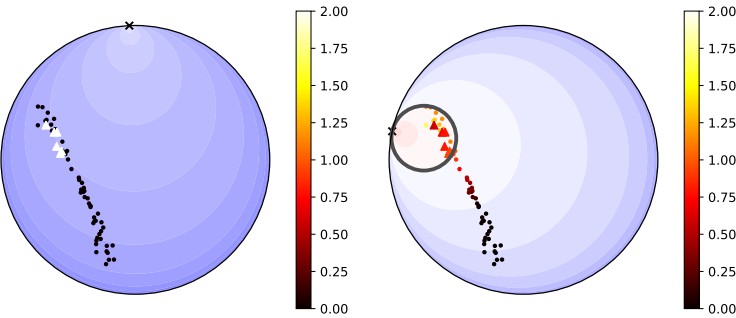

Figure 3: **Importance of class balancing.** The optimal solution on a binary classification problem found (left) without and (right) with class-balancing. The left horosphere has a large negative **b** and does not split the data. In contrast, the right horosphere has a higher loss but a positive information gain. Class labels are given by marker shape. The points and color bar are colored by loss; the background is colored by distance to the horosphere.

where $\gamma_i = \frac{1}{\sqrt{1 - \|\mathbf{x}_i\|^2}}$. Afterward, the mean is transformed back to the Poincaré ball with

$$\mathbf{x}_\mathbb{B} = \frac{\mathbf{x}_\mathbb{K}}{1 + \sqrt{1 - \|\mathbf{x}_\mathbb{K}\|^2}}. \tag{11}$$

We compute all pairwise similarities and iteratively group the two most similar classes together, where the similarity is based on the LCA of the means of the two classes. The LCA for two points in hyperbolic space is the point that is closest to the origin on the geodesic between them (Chami et al., 2020). As a result, the distance of the LCA from the origin can be seen as a similarity measure. Given two means $\boldsymbol{\phi}_i$ and $\boldsymbol{\phi}_j$, we can calculate their similarity by

$$\text{sim}(\boldsymbol{\phi}_i, \boldsymbol{\phi}_j) = 2 \tanh^{-1}\left(\sqrt{R^2 + 1} - R\right) \tag{12}$$

with

$$R = \sqrt{\left(\frac{\|\boldsymbol{\phi}_i\|^2 + 1}{2\|\boldsymbol{\phi}_i\| \cos(\alpha)}\right)^2 - 1} \tag{13}$$

and

$$\alpha = \tan^{-1}\left(\frac{1}{\sin(\theta)}\left(\frac{\|\boldsymbol{\phi}_i\|(\|\boldsymbol{\phi}_j\|^2 + 1)}{\|\boldsymbol{\phi}_j\|(\|\boldsymbol{\phi}_i\|^2 + 1)} - \cos(\theta)\right)\right), \tag{14}$$

where $\theta$ is the angle between $\boldsymbol{\phi}_i$ and $\boldsymbol{\phi}_j$. We repeat this until only two hyperclasses are left, for $K - 2$ iterations total. As a result, we only evaluate $K + K - 2$ horospheres per node. We show an example where evaluating hyperclasses is beneficial in Fig. 2.

### 3.2.4 Imbalanced data

SVM-based methods are known to behave poorly on imbalanced data, and `HoroSplitter` is no exception. We address this by proposing a class-balanced version of the `HoroSplitter`. Specifically, the problem comes from equation 7 finding an optimal solution that does not split the data and has an information gain of zero as a result. As an example, consider the binary problem in Figure 3. The optimal horosphere has its ideal point on the top of the Poincaré ball, a $\mu$ very close to 0 and $o$ very close to 1; a result of the loss, in the case of a large imbalance, almost reducing to the degenerate form

$$\ell_{\text{imbalanced}}(\mu, \mathbf{w}, o; D) \cong C \sum_{i=1}^{N} \max(0, 1 + y_i). \tag{15}$$

We remedy this limitation by incorporating class-balancing into `HoroSplitter`'s loss function (Cui et al., 2019). We rewrite the loss function in equation 7 as

$$\boldsymbol{\ell}(\mu, \mathbf{w}, o; D) = \frac{1}{2}\mu^2 + C \sum_{i=1}^{N} \frac{1-\beta}{1-\beta^{n_y}} \max(0, 1 - y_i(\mu B_{\mathbf{w}}^{-1}(\mathbf{x}_i) - o)), \tag{16}$$

where $n_y$ gives the number of training samples belonging to class $y$ and $\beta$ is a hyperparameter. By adding more emphasis on the loss for samples of the minority class, we can find horospheres that split the data even in imbalanced cases.

**Training and inference**   We build horospherical decision trees (HoroDT) by repeatedly applying the `HoroSplitter`. This process is repeated until all nodes are pure, *i.e.* contain one class, or a stopping criterion is met. Example stopping criteria include stopping when a node reaches a certain number of samples or no split with an information gain higher than a threshold can be found. By combining multiple HoroDTs into an ensemble, we form a horospherical Random Forest (HoroRF). Each tree is trained on a randomly sampled (with replacement) subset of the data, and a subsample of the features is considered at every split. Predictions are made by a majority vote among the HoroDTs.

## 4   Experiments

### 4.1   Datasets

We evaluate HoroRF on two canonical hyperbolic classification benchmarks: WordNet subtree and network node classification (Cho et al., 2019; Fan et al., 2023). Additionally, we introduce three new multi-class WordNet experiments, which we will detail first.

**Multi-class WordNet**   We propose three new multi-class WordNet experiments aimed at evaluating three distinct qualities. In the first experiment, the task is to classify samples into one of multiple subtrees with the same parent. In the second experiment, we pick nested subtrees, and samples must be classified into the smallest subtree they belong to. The third experiment combines the previous experiments with nested classes of multiple subtrees. Unlike previous benchmarks, these are difficult multi-class experiments on large, imbalanced datasets. We provide the full details of the experiments in the appendix. The datasets are split into train, validation, and test sets with a ratio of 60:20:20. A hyperparameter search is done on the validation set, and the macro-f1 score over three trials on the test set is reported.

**WordNet**   We run binary classification experiments on the nouns of WordNet (Fellbaum, 2010), embedded into hyperbolic space using hyperbolic entailment cones (Ganea et al., 2018a). The goal is to identify whether a sample belongs to a semantic category (*i.e.*, a subtree of the hyperbolic embeddings). Following (Fan et al., 2023), we split the data into 80% for training and 20% for testing and report the best results from a grid search over three runs in AUPR. We find that three of the larger subtrees typically used for these experiments (`animal.n.01`, `group.n.01`, `mammal.n.01`) can be solved near-perfectly with random forests, both Euclidean and hyperbolic. Therefore, we add experiments on two additional subtrees, `occupation.n.01` and `rodent.n.01`, which are smaller and more difficult subtrees for classification purposes.

**Networks**   We run node classification experiments on four network datasets embedded into hyperbolic space. These networks are (1) *karate* (Zachary, 1977), a network of 34 members of a karate club split into two factions, (2) *polblogs* (Adamic & Glance, 2005), a network of 1224 hyperlinks in political blogs split into two communities, (3) football (Girvan & Newman, 2002), a network of 115 colleges linked by football games split into 12 leagues, and (4) *polbooks*[1], a network of 105 books split into three affiliations.

We use the public embeddings provided by Cho et al. (2019). Each dataset has five sets of embeddings obtained using the method of Chamberlain et al. (2017). Following previous work (Fan et al., 2023), we run a grid search with 5-fold stratified cross-validation and report the best average micro-f1 score over five seeds, using a different embedding for each seed.

---

[1]http://www-personal.umich.edu/ mejn/netdata/

## 4.2   Evaluation

**Baselines**   We evaluate HoroRF against six baselines, split into two categories. The first set covers hyperbolic classifier baselines, namely hyperbolic multiple logistic regression (HypMLR; Ganea et al. (2018b)) and the state-of-the-art hyperbolic SVM, HoroSVM (Fan et al., 2023). The second set of baselines comprises Euclidean counterparts of the hyperbolic classifiers, namely a linear SVM (LinSVM), an SVM with an RBF kernel (RBFSVM; (Cortes & Vapnik, 1995)), an axis-aligned random forest (RF) (Breiman, 2001), and an oblique random forest (OblRF) with a similar set-up to HoroRF. The OblRF uses a cost-balanced linear SVM to find optimal splits (see *e.g.* (Do et al., 2010)). If the SVM fails to converge, it defaults to using the best axis-aligned split. It finds hyperclasses by iteratively merging the two classes with the closest means.

As is common practice (Cho et al., 2019; Fan et al., 2023), the Euclidean baselines are run on the same embeddings as the hyperbolic methods, which we also find to perform better compared to mapping the embeddings back to Euclidean space first. Moreover, as the SVMs, HypMLR, and HoroSVM baselines are negatively affected by the class imbalance in the WordNet experiments, we report cost-balanced results on those benchmarks, where the loss is weighted inversely proportional to the class frequency.

**Implementation Details**   We use 100 trees for all tree-based methods. We use the Gini impurity to calculate the information gain. At every node, we set $C$ to $2^n$ where $n$ is randomly sampled from $\{-3, -2, \ldots, 5\}$. We stop splitting when a node reaches a certain number of samples $m$. In the case of ties in information gain between possible splits, a random split is selected. If the `HoroSplitter` fails to converge, we choose the best horosphere from a small number of random ideal points. We set $m$ and the class-balancing hyperparameter $\beta$ via grid search. Further details are given in the appendix.

## 4.3   Results

**WordNet**   The results for the WordNet subtree classification experiments are shown in Tab. 1. The Euclidean SVMs are unable to handle the large imbalance in the datasets despite the loss re-weighting. The random forest approaches are better equipped to deal with this type of data. The results show that HoroRF outperforms the Euclidean tree-based methods in most cases. The differences are most pronounced in the hardest experiments, *occupation* and *rodent*, while we find the tree-based methods to perform similarly on the saturated experiments with close to perfect scores, i.e., *animal*, *group*, and *mammal*. Overall, HoroRF performs better than the well-known Euclidean SVMs and random forest approaches.

Compared to the hyperbolic classifiers, we find that both the hyperbolic MLR and SVM baselines struggle to deal with the imbalanced data, despite the loss re-weighting. Neither are able to match the performance of HoroRF. We argue this is the case as real-world data cannot be distinguished with a single horosphere, but requires recursive horospherical separation as done in HoroRF. We conclude from the WordNet experiment that HorRF is a competitive classifier even in the binary setting.

**Multi-class WordNet**   The results for the multi-class WordNet experiments are shown in Table 2, which paint a similar picture as the previous experiments. For the Euclidean methods, the tree-based methods outperform the SVMs, with OblRF, on average, having a slight edge over the axis-aligned random forest. HoroRF outperforms the SVMs and axis-aligned RF in all three cases, and despite reaching an equal score in the first experiment, it outperforms OblRF in the remaining two. Compared to the other hyperbolic classifiers, HoroRF reaches far higher performance on these complex imbalanced multi-class experiments, on which both HypMLR and HoroSVM struggle greatly.

**Networks**   The results on the network datasets are shown in Table 3. Our HoroRF outperforms the Euclidean methods in all cases, besides matching the SVMs on *karate*, showing that the choice of horospheres over hyperplanes results in better performance on data lying in hyperbolic space.

As for the hyperbolic methods, HypMLR is outclassed by both HoroRF and HoroSVM in all cases except *football*. HoroSVM and HoroRF perform similarly in the binary experiments, with a slight edge for HoroRF on *polblogs*. In contrast, for the multi-class experiments *football* and *polbooks*, HoroRF outclasses HoroSVM

| | Animal | Group | Worker | Mammal | Tree | Solid | Occupation | Rodent |
|---|---|---|---|---|---|---|---|---|
| **Euclidean** | | | | | | | | |
| LinSVM | 2.5±0.0 | 6.0±0.6 | 0.7±0.0 | 0.7±0.0 | 0.6±0.0 | 0.8±0.0 | 0.2±0.0 | 0.1±0.0 |
| RBFSVM | 2.5±0.0 | 6.0±0.6 | 0.7±0.0 | 0.7±0.0 | 0.6±0.0 | 0.8±0.0 | 0.2±0.0 | 0.1±0.0 |
| RF | **98.6**±0.2 | 96.6±0.4 | 70.0±3.1 | **99.3**±0.3 | 75.8±2.0 | 92.5± 1.2 | 59.8±1.1 | 34.5±3.0 |
| OblRF | 98.4±0.1 | **96.7** ±0.3 | 73.3±1.3 | 98.8±0.3 | 76.2±3.2 | 92.9±1.7 | 65.0±2.4 | 38.2±0.9 |
| **Hyperbolic** | | | | | | | | |
| HypMLR | 3.2 ±0.1 | 6.5 ±0.0 | 1.2 ±0.2 | 0.9 ±0.0 | 1.3±0.8 | 1.0 ±0.2 | 0.9±0.9 | 0.4 ±0.4 |
| HoroSVM | 92.4±1.3 | 71.0±0.5 | 39.9±3.6 | 89.8±0.9 | 41.5±1.0 | 65.7±0.9 | 13.0±1.5 | 13.6±2.0 |
| *HoroRF* | 98.5±0.2 | 96.6±0.4 | **73.4**± 1.7 | **99.3**±0.4 | **76.6**± 3.8 | **93.5**± 1.3 | **65.6**± 1.5 | **39.0**± 2.4 |

Table 1: **Comparative evaluation on binary WordNet experiments.** We follow the experimental protocol of Fan et al. (2023) and report the mean and standard deviation of the AUPR over three runs. Underlined gives the best hyperbolic method, **Bold** denotes the best method overall. On aggregate across the experiments, HoroRF obtains the best performance.

| | Same level | Nested | Both |
|---|---|---|---|
| **Euclidean** | | | |
| LinSVM | 48.8 ±0.9 | 59.7 ±0.5 | 35.6 ±0.2 |
| RBFSVM | 80.0 ±1.8 | 89.8 ±1.1 | 70.8 ±0.6 |
| RF | 89.7±0.7 | 91.7±0.3 | 81.5±1.5 |
| OblRF | **91.3**±0.6 | 93.0±0.2 | 81.3±1.4 |
| **Hyperbolic** | | | |
| HypMLR | 29.0 ±4.5 | 57.6 ±0.5 | 35.9 ±6.7 |
| HoroSVM | 50.2 ±2.2 | 56.0 ±0.7 | 35.2 ±0.5 |
| *HoroRF* | **91.3**±0.3 | **93.3**±1.1 | **81.9**±1.5 |

Table 2: **Comparative evaluation on multi-class WordNet.** We run a grid search to determine the optimal configuration on the validation set and report results over three trials in macro-f1 score on the test set. Underlined gives the best hyperbolic method, **Bold** denotes the best method overall. HoroRF performs best in all settings.

by 4.0 and 0.6 macro f-1, respectively, confirming the benefit of HoroRF over HoroSVM in more complex cases. Overall, we find that HoroRF is the most consistent and best-performing classifier across all benchmarks.

## 4.4 Ablations and Visualizations

**Hierarchical Classification** Hyperbolic methods have been proven successful when evaluated on a wide range of hierarchical tasks Cao et al. (2020); Ghadimi Atigh et al. (2021); Surís et al. (2021); Tifrea et al. (2019). Here, we aim to verify our hypothesis that random forests are well-suited for hyperbolic space by comparing HoroRF with other SOTA hyperbolic classifiers on hierarchical classification, with metrics explicitly designed to evaluate their success in capturing the hierarchy.

We evaluate the methods using CIFAR10 Krizhevsky et al. (2009) and STL10 Coates et al. (2011) using the CIFAR10 hierarchy from Sebastian & Sebastian (2023). Note that STL10 has the same classes as CIFAR10, with the only difference being the substitution of `frog` for `monkey`. As such, we maintain the hierarchy from Sebastian & Sebastian (2023) for STL10 but modify it by removing `frog` and incorporating `monkey` into the mammal subtree.

We embed the images into 9-dimensional space by training a hyperbolic prototype ResNet-18 with prototypes uniformly distributed over the hyperbole Kasarla et al. (2022) and train the classifiers on these 9-dimensional

|  | Binary | | Multi-class | |
|---|---|---|---|---|
|  | Karate | Polbooks | Football | Polblogs |
| **Euclidean** | | | | |
| LinSVM | **95.4**±2.3 | 92.4±0.3 | 33.2±5.1 | 85.5±0.9 |
| RBFSVM | **95.4**±2.3 | 92.4±0.3 | 35.5±4.7 | 84.4±1.9 |
| RF | 94.3±3.1 | 92.1±0.3 | 36.2±4.9 | 85.1±2.1 |
| OblRF | 94.8 ±2.2 | 92.1 ±0.3 | 36.7 ±2.7 | 84.4 ±1.3 |
| **Hyperbolic** | | | | |
| HypMLR | 93.1±3.4 | 90.9±0.7 | **40.2**±2.5 | 81.5±1.5 |
| HoroSVM | **95.4**±2.3 | 92.4±0.2 | 34.3±1.8 | 85.3±0.8 |
| *HoroRF* | **95.4**±2.3 | **92.5**±0.3 | 38.3±1.8 | **86.1**±1.0 |

Table 3: **Comparative evaluation on network datasets.** We follow the experimental protocol of Fan et al. (2023) and report the mean and standard deviation of the micro-f1 score over five trials of 5-fold stratified cross-validation. Underlined gives the best hyperbolic method, **Bold** denotes the best method overall. HoroRF obtains the best performance on aggregate over the datasets.

|  | CIFAR10 | | STL10 | |
|---|---|---|---|---|
|  | Mis. Sev. (↓) | HD@1 (↓) | Mis. Sev. (↓) | HD@1 (↓) |
| HypMLR | 60.8±1.7 | 10.2±0.0 | 54.2±2.8 | 22.4±0.3 |
| HoroSVM | 59.9±0.1 | 12.5±0.0 | 54.4±1.0 | 26.0±0.3 |
| *HoroRF* | **58.2**±0.3 | **9.9**±0.2 | **53.4**±0.2 | **22.3**±0.1 |

Table 4: **Evaluating hyperbolic classifiers for hierarchical classification.** We run a grid search to determine the optimal configuration on the validation set and report the mean and standard deviation of the mistake severity and hierarchical distance@1 on the test set over three seeds. **Bold** denotes the best method. HoroRF performs best.

image features. We obtain hyperbolic embeddings for 3600 training, 900 validation, and 1000 test samples. To compare the classifiers, we run a grid search on the training set and select the optimal configuration based on validation performance. Then, we report the mistake severity and hierarchical distance@1 on the test set Garg et al. (2022). These metrics both use the LCA distance, computed by dividing the height of the LCA of the predicted and ground-truth class by the tree height. The mistake severity is the average of this distance for all misclassified samples, whereas the HD@1 computes the average distance for all the samples Karthik et al. (2021).

From the results in Tab. 4, HoroRF is the best hyperbolic method, confirming its ability to model the hierarchy effectively and the benefits of our generalization of random forests to hyperbolic space.

**Ablating hyperclasses and balancing.** We perform an ablation study to validate our choices on the *football* dataset. From Tab. 5, we find a large benefit in using our `HoroSplitter` to find splits instead of enumerating all possible horospheres at axis-aligned ideal points. Furthermore, we show how both our hyperclass and class-balancing additions improve upon the base `HoroSplitter`, and their combination enables us to reach the best performance.

**Visualization & synthetic evaluation** We build a synthetic dataset following Ganea et al. (2018a) by creating a synthetic tree of depth six with a branching factor of four and embedding it into hyperbolic space with hyperbolic entailment cones. We then partition the nodes into five classes from two depths. Using this dataset, we visualize the splits of HoroRF, HoroRF without hyperclasses or class-balancing, OblRF on the hyperbolic embeddings, and OblRF on the Euclidean embeddings in Fig. 6. We additionally verify which

| Ideal Points | Hyper-classes | Class-balanced | f1 |
|---|---|---|---|
| Axis-aligned | ✗ | ✗ | 9.9 |
| `HoroSplitter` | ✗ | ✗ | 35.7 |
| `HoroSplitter` | ✗ | ✓ | 36.5 |
| `HoroSplitter` | ✓ | ✗ | 37.4 |
| `HoroSplitter` | ✓ | ✓ | **38.3** |

Table 5: **Ablating the `HoroSplitter`** in HoroRF on the *football* dataset. We follow the experimental protocol of Fan et al. (2023) and report the mean and standard deviation of the micro-f1 score over five trials of 5-fold stratified cross-validation. Our splitting function, hyperclass-based aggregation, and class balancing are all important for effective hyperbolic random forests.

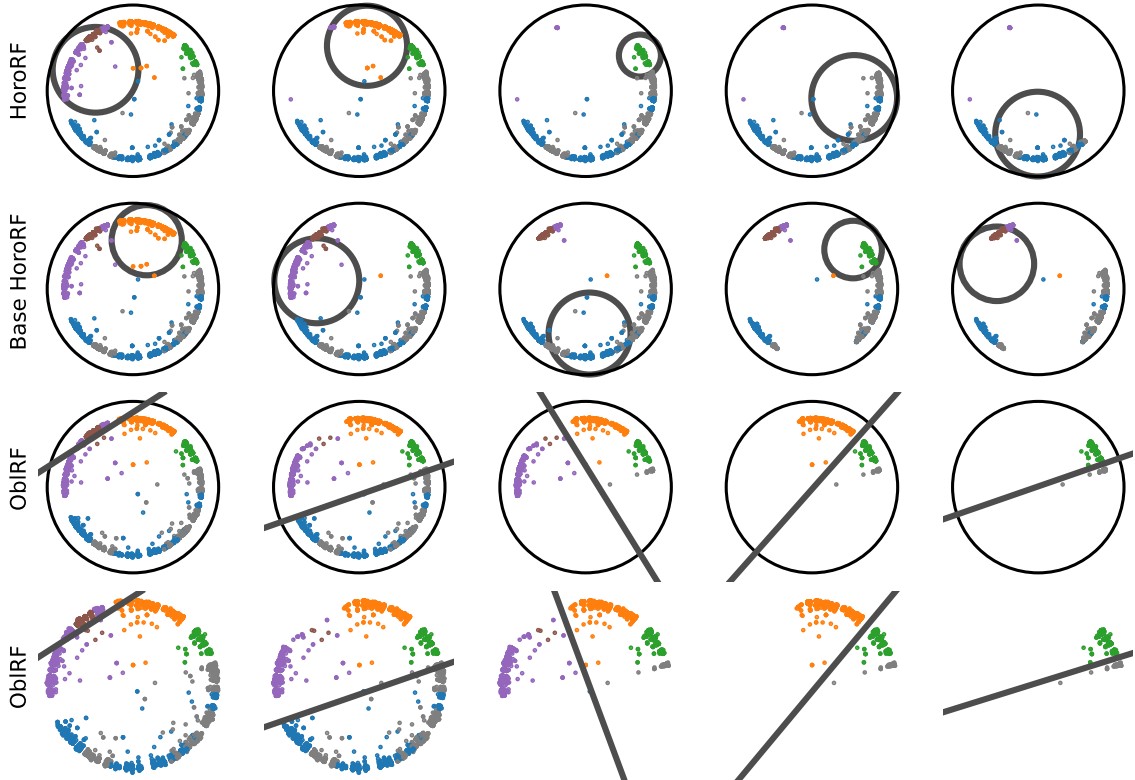

Figure 4: **Visualizing HoroRF & OblRF splits.** We show five splits for HoroRF, a basic version of HoroRF without class-balancing and hyperclasses, OblRF on the original hyperbolic embeddings, and OblRF after mapping the data to Euclidean space. The split with the highest impurity is chosen at every level to visualize the next level. Splitting in hyperbolic space with balanced and multi-level horospheres obtains good splits for hyperbolic classification.

method is the most successful at various depths in Tab. 6, where we find that HoroRF consistently reaches high performance and is especially effective with a single split.

**Reducing runtime**  A limitation of HoroRF is its runtime on large datasets. While the `HoroSplitter` scales linearly with the number of samples as HoroSVM has a linear time complexity, it is applied multiple times per node and repeated for every node. We designed and conducted two experiments showing how to mitigate the computational time needed to obtain good results: reducing the number of trees and lowering

| Depth | 1 | 2 | 3 | 10 | No limit |
|---|---|---|---|---|---|
| OblRF Euc | 34.5±0.1 | **68.8**±2.5 | 83.3±0.0 | **99.3**±0.0 | 98.9±0.0 |
| OblRF Hyp | 33.6±0.7 | 65.6±1.3 | 83.2±0.3 | 99.1±0.0 | 98.9±0.0 |
| Base HoroRF | 60.8±0.0 | 67.8±4.9 | 83.6±1.6 | 99.1±0.0 | **99.1**±0.3 |
| HoroRF | **63.1**±3.2 | 67.4±5.1 | **85.5**±1.8 | **99.3**±0.0 | 98.9±0.3 |

Table 6: **Effect of tree depth on performance on the synthetic dataset.** We run a grid search to determine the optimal configuration on the validation set and report results over three trials in micro-f1 score on the test set. **Bold** denotes the best method, underlined the second best. Splitting in hyperbolic space with balanced and multi-level horospheres consistently reaches high performance.

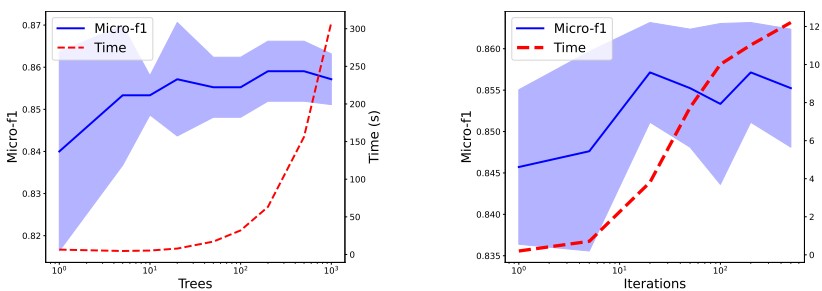

Figure 5: **Analyzing ways to reduce the computational time of HoroRF on *polbooks*.** In (a), we show that the optimal performance is already reached with around 20 trees. In (b), we show that by reducing the number of optimization iterations of HoroSVM, we can further reduce computational time while maintaining high performance.

the number of HoroSVM optimization iterations. From the results in Fig. 5, we find that both the number of trees and HoroSVM optimization iterations can be greatly reduced without sacrificing performance. Lowering the number of HoroSVM iterations is a logical way to reduce computational time, as we are not interested in finding the exact optimal horosphere; a solution close to it suffices for our purpose and might even benefit HoroRF due to the increased variation in splits.

## 5   Conclusion

We present HoroRF, a random forest algorithm in hyperbolic space. HoroRF constructs its trees by repeated application of the `HoroSplitter`, which finds horosphere-based splits with high information gain. We show the benefits of additional components aimed at improving performance on imbalanced and multi-class datasets. Extensive experiments on two established and one newly introduced benchmark show its effectiveness over both existing hyperbolic classifiers as well as their Euclidean counterparts.

Further advances in hyperbolic classification can easily be incorporated into the framework, as HoroRF is in no way restricted to a single type of classifier. In future work, we aim to investigate the benefits of combining multiple types of splits, for example by including hyperbolic logistic regression to provide additional split options to include next to the horosphere splits. We furthermore expect that any improvement in optimization of HoroSVM as splitting operation, akin to hyper-optimized libraries for Euclidean SVMs (Chang & Lin, 2011), will have direct benefits for HoroRF, especially when dealing with many classes.

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
