

Figure 6: **Visualizing the multi-class WordNet experiments.** We embed all WordNet nouns into two dimensions with hyperbolic entailment cones Ganea et al. (2018a) and color the different classes of our three multi-class WordNet experiments.

# 6 Supplementary Material

## 6.1 Dataset details

**Multi-class WordNet** For the first experiment, we take four subtrees of the `vertebrate.n.01` subtree: `reptile.n.01` (237 samples), `mammal.n.01` (944 samples), `bird.n.01` (697 samples), and `aquatic_vertebrate.n.01` (506 samples). Together with the negative class of 63307 samples, this leads to a 5-way classification problem.

For the second experiment, we take nested subtrees: `physical_entity.n.01` (21252 samples), `living_thing.n.01` (12460 samples), `animal.n.01` (2268 samples), `mammal.n.01` (767 samples), and `canine.n.01` (178 samples). Together with the negative class of 28766 samples, this leads to a 6-way classification problem.

For the third experiment, we combine the two. We take the `act.n.01` subtree (4182 samples), as well as its subtree `happening.n.01` (639 samples). Additionally, we include `action.n.01` (1317 samples) and its subtree `movement.n.03` (141 samples). Together with the negative class of 59412 samples, this leads to a 5-way classification problem.

Visualizations in 2D for the three experiments are given in Fig. 6. Due to the multi-class setting and large class imbalance in all experiments, we use the macro-f1 to evaluate methods.

**WordNet** The 82115 nodes of the WordNet noun hierarchy are split into two classes for each experiment. The corresponding number of samples in the train and test split for the negative and positive classes are given in Tab. 7. Due to the large class imbalance in all experiments, we use the AUPR to evaluate methods.

|  | Animal | Group | Worker | Mammal | Tree | Solid | Occupation | Rodent |
|---|---|---|---|---|---|---|---|---|
| **Train** | | | | | | | | |
| Positive | 3213 | 6701 | 892 | 945 | 811 | 985 | 226 | 110 |
| Negative | 62478 | 58990 | 64799 | 64746 | 64880 | 64706 | 65465 | 65581 |
| **Test** | | | | | | | | |
| Positive | 803 | 1675 | 223 | 236 | 203 | 246 | 57 | 28 |
| Negative | 15620 | 14748 | 16200 | 16187 | 16220 | 16177 | 16366 | 16395 |

Table 7: **WordNet dataset details.** We show the class distribution for the train and test split per experiment.

|  | *an* | *oms* | *pt* |
|---|---|---|---|
| **Euclidean** | | | |
| LinSVM | 100±0.0 | 91.4±1.1 | 41.2±0.0 |
| RBFSVM | 100±0.0 | 92.8±0.0 | 43.0±0.0 |
| RF | 97.8±1.6 | 91.3±0.2 | 47.7±0.8 |
| OblRF | 100±0.0 | 92.3±0.2 | 42.4±0.9 |
| **Hyperbolic** | | | |
| *HoroRF* | 100±0.0 | 90.5±0.4 | 44.0±1.1 |

Table 8: **Comparative evaluation on tabular experiments.** We run a grid search to determine the optimal configuration on the validation set and report the mean and standard deviation of the micro f1-score on the test set averaged over three runs. HoroRF obtains competitive performance.

**Networks**  As described in the main text, the networks are (1) *karate* (Zachary, 1977), with 34 nodes and 2 classes, (2) *polblogs* (Adamic & Glance, 2005), with 1224 nodes and 2 classes, (3) football (Girvan & Newman, 2002), with 115 nodes and 12 classes, and (4) *polbooks*[2], with 105 nodes and 3 classes.

### 6.2   Full implementation details

We run a grid search for all methods. For the linear SVM, we search over $C \in \{-5, -4, \cdots, 15\}$. For the RBF SVM, we search over $C \in \{-5, -1, \cdots, 15\}$ and $\gamma \in \{-15, -11, \cdots, 1, 3\}$. For the random forest and oblique random forest, we search over the hyperparameter that controls at what number of samples to stop splitting a node $m \in \{1, 3, 5\}$ on the small network datasets and $m \in \{3, 7, 11\}$ on the larger WordNet datasets. For the hyperbolic logistic regression, we search over the learning rate $lr \in \{0.1, 0.01, 0.001\}$, and the batch size $bs \in \{16, 32, 64\}$ on the network datasets and $bs \in \{128, 256, 512\}$ on the WordNet datasets. For HoroSVM, we search over $C \in \{-5, -4, \cdots, 15\}$. For HoroRF, we search over $\beta \in \{0, 0.9, \cdots, 0.9999\}$, and $m \in \{1, 3, 5\}$ on the network datasets and $m \in \{3, 7, 11\}$ on the WordNet datasets.

We run all our experiments on a machine running CentOS 7.9.2009 with an AMD EPYC 7302 16-Core Processor with access to 48GB of memory. We set our seeds such that our implementation of the HoroSVM baseline resembles the results on the network datasets reported in its original publication.

### 6.3   Tabular data

We aim to investigate how our method works on real-world tabular data. We compare HoroRF against the Euclidean classifiers on experiments 1-3 of the UCI121 benchmark (acute-nephritis, oocytes-merluccius-states-2f, and primary-tumor) Fernández-Delgado et al. (2014), using the preprocessed versions of Cai et al. (2021). We find that mapping these Euclidean datasets to the Poincaré ball with a curvature of 1 leads to all samples lying on the boundary. For this reason, we also search over the curvature parameter in $c \in [1, 0.1, 0.01]$. We show results on the datasets in Tab. 8, finding that HoroRF is competitive with the Euclidean classifiers despite being designed for a different data representation.

### 6.4   Different embeddings

**Euclidean embeddings**  All results in the main paper are obtained by fitting the methods to the original hyperbolic embeddings. In Tabs. 9- 11, we show that applying the Euclidean methods to the original hyperbolic embeddings outperforms applying them to the embeddings mapped to Euclidean space.

**Hyperboloid embeddings & comparison to HyperRF**  The concurrent work of Chlenski et al. (2023) finds splits as midpoints between angles of data points in the hyperboloid model of hyperbolic space, rather

---

[2]http://www-personal.umich.edu/ mejn/netdata/

| | Animal | Group | Worker | Mammal | Tree | Solid | Occupation | Rodent |
|---|---|---|---|---|---|---|---|---|
| **Euc. embeddings** | | | | | | | | |
| LinSVM | 2.5±0.0 | 5.6±0.1 | 0.7±0.0 | 0.7±0.0 | 0.6±0.0 | 0.8±0.0 | 0.2±0.0 | 0.1±0.0 |
| RBFSVM | 2.5±0.0 | 5.4±0.0 | 0.7±0.0 | 0.7±0.0 | 0.6±0.0 | 0.8±0.0 | 0.2±0.0 | 0.1±0.0 |
| RF | 98.5±0.2 | 96.4±0.1 | 72.5±1.0 | 98.4±1.0 | **77.2**±2.3 | 91.2±0.8 | 63.4±0.6 | 28.6±6.5 |
| OblRF | 98.1±0.1 | 96.2±0.2 | 71.4±0.8 | 98.8±0.7 | 77.0±1.6 | 91.4±0.7 | 62.9±0.3 | 29.3±7.4 |
| **Hyp. embeddings** | | | | | | | | |
| LinSVM | 2.5±0.0 | 6.0±0.6 | 0.7±0.0 | 0.7±0.0 | 0.6±0.0 | 0.8±0.0 | 0.2±0.0 | 0.1±0.0 |
| RBFSVM | 2.5±0.0 | 6.0±0.6 | 0.7±0.0 | 0.7±0.0 | 0.6±0.0 | 0.8±0.0 | 0.2±0.0 | 0.1±0.0 |
| RF | **98.6**±0.2 | 96.6±0.4 | 70.0±3.1 | **99.3**±0.3 | 75.8±2.0 | 92.5± 1.2 | 59.8±1.1 | 34.5±3.0 |
| OblRF | 98.4±0.1 | **96.7** ±0.3 | **73.3**±1.3 | 98.8±0.3 | 76.2±3.2 | **92.9**±1.7 | **65.0**±2.4 | **38.2**±0.9 |

Table 9: **Comparing Euclidean methods applied to different embeddings on binary WordNet experiments.** We follow the experimental protocol of Fan et al. (2023). **Bold** denotes the best method. Euclidean classifiers perform better on the hyperbolic embeddings.

| | Same level | Nested | Both |
|---|---|---|---|
| **Euc. embeddings** | | | |
| LinSVM | 47.7±1.3 | 51.0±0.3 | 32.9±10.2 |
| RBFSVM | 81.3±2.1 | 89.8±1.0 | 73.8±0.6 |
| RF | 88.9±1.0 | 92.0±0.4 | 81.1±0.4 |
| OblRF | 90.5±0.7 | 92.2±0.2 | **82.0**±0.7 |
| **Hyp. embeddings** | | | |
| LinSVM | 48.8±0.9 | 59.7±0.5 | 35.6±0.2 |
| RBFSVM | 80.0±1.8 | 89.8±1.1 | 70.8±0.6 |
| RF | 89.7±0.7 | 91.7±0.3 | 81.5±1.5 |
| OblRF | **91.3**±0.6 | **93.0**±0.2 | 81.3±1.4 |

Table 10: **Comparing Euclidean methods applied to different embeddings on multi-class Word-Net experiments.** We follow our experimental protocol as described in the main paper. **Bold** denotes the best method. Euclidean classifiers perform better on the hyperbolic embeddings.

| | Binary | | Multi-class | |
|---|---|---|---|---|
| | Karate | Polbooks | Football | Polblogs |
| **Euc. embeddings** | | | | |
| LinSVM | **95.4**±2.3 | 92.4±0.3 | 33.4±5.2 | 85.7±0.6 |
| RBFSVM | **95.4**±2.3 | **92.5**±0.2 | 35.1±3.3 | 83.6±1.5 |
| RF | 94.3±3.1 | 92.1±0.3 | 35.0±5.1 | 84.8±1.9 |
| OblRF | 94.9±3.3 | 92.0±0.6 | 35.3±2.5 | 83.8±1.2 |
| **Hyp. embeddings** | | | | |
| LinSVM | **95.4**±2.3 | 92.4±0.3 | 33.2±5.1 | 85.5±0.9 |
| RBFSVM | **95.4**±2.3 | 92.4±0.3 | 35.5±4.7 | 84.4±1.9 |
| RF | 94.3±3.1 | 92.1±0.3 | 36.2±4.9 | 85.1±2.1 |
| OblRF | 94.8±2.2 | 92.1±0.3 | **36.7**±2.7 | 84.4±1.3 |

Table 11: **Comparing Euclidean methods applied to different embeddings on network datasets.** We follow the experimental protocol of Fan et al. (2023). **Bold** denotes the best method. Results are similar between embeddings.

|         | Football | Polblogs | Worker | Tree | Occupation | Rodent | Same level | Nested | Mean |
|---------|----------|----------|--------|------|------------|--------|------------|--------|------|
| HyperRF | 33.0±2.3 | 84.4±1.8 | 71.5± 0.9 | **77.5**± 2.1 | 53.9± 4.4 | **40.3**± 1.6 | **91.7**±1.0 | **93.8** ±0.8 | 68.3 |
| *HoroRF* | **38.3**±1.8 | **86.1**±1.0 | **73.4**± 1.7 | 76.6± 3.8 | **65.6**± 1.5 | 39.0± 2.4 | 91.3±0.3 | 93.3±1.1 | **70.4** |

Table 12: **Comparative evaluation between HyperRF and HoroRF.** We follow the experimental protocol for each experiment as described in the main text. **Bold** denotes the best method. The methods are comparable across the benchmarks.

than the Poincaré ball model used in our work. We compare HoroRF to HyperRF on the multi-class networks, the four most imbalanced WordNet experiments, and the same-level and nested multi-class experiments in Tab. 12. We find that HoroRF outperforms HyperRF on the networks, they perform similarly on the WordNet subtree classification, and HyperRF slightly outperforms HoroRF on the multi-class WordNet experiments.