# OpenReview forum: "Hyperbolic Random Forests"
_TMLR — Accepted by TMLR_

### Review · Reviewer_H8RP · 2024-03-03

**Summary Of Contributions:**

The paper describes an implementation of a random forest classifier in hyperbolic space. The implementation seems to be based closely on the Euclidean oblique random forest (OblRF) with SVM of Do et al. 2010 (reference from the paper), but with the SVM component modified to the hyperbolic SVM. Some heuristic modifications on how to perform individual splits, such as aggregating classes and modifying the loss function to accommodate class imbalance are discussed.

Some new and existing experimental set ups for hyperbolic data classification and associated performance comparisons are presented, showing some potential of the proposed approach.

**Audience:**

Yes

**Broader Impact Concerns:**

Nothing immediately apparent to note

**Claims And Evidence:**

No

**Requested Changes:**

Questions and requested changes:
1. Is the class merging heuristic used in the paper novel, or at least very similar to something already in use? If not, please reference its origin.
2. Were hyperparameters chosen to optimise test error, or were they selected by an appropriate data driven selection technique like cross validation (CV)? If the former, then I am of the opinion that the results are meaningless since they do not reflect what is realisable in practice, and I'd like to request results be reported from experiments where hyperparameters are selected using more principled approaches which do not rely on access to test error. Please also provide results from more than three replications since the improvements offered by the proposed approach are not very substantial and stronger evidence is needed to show the changes proposed are worthwhile.
3. Since there is another recent implementation of random forests in hyperbolic space, referenced in the paper, please provide comparisons with this method as well as they seem arguably the most relevant.
4. There are numerous typographical errors in the paper, some of which I will list below, along with some other minor points. Please proof-read before uploading a revision:
- In reference to Fig 1. c), you refer to a "subtree". This is a bit of a leap for any reader not very clear on decision tree method estimation algorithms, I'd recommend something more descriptive of the figure itself.
- In Fig 1. I would argue that in a) the linear model actually does a good job. Is the problem that it would need more splits to get to the green class? Please clarify.
- In Eq. (4) the argument of B_w should be bold font. In addition, in the following sentence should w be in S^{n-1}?
- In Eq. (6) the subscript in B_w should be bold font (or at least be consistent across the paper)
- The alignment in Eq. (7) is rather unsightly.
- Also in Eq. (7), why is there a bracket around B^{-1}_w(x)? The way it appears \mu looks like it must be a function.
- Below Eq. (8) C should be in math font.
- "Eq. equation 7" shouldn't have both "Eq." and "equation" (numerous instances of this)
- In Eq. (16) should the \beta^{n_y} not be for the specific class of instance i? Otherwise this term is the same for all i and does nothing.
- Regarding the above, I would recommend not going into the rather rough explanation of the effect of class imbalance and just jump right to Eq. (16) and say the extra factor is to accommodate unbalanced classes.

**Strengths And Weaknesses:**

The increased interest in hyperbolic representations of data structure and associated algorithms and models for their analysis places the paper in a relevant context. Furthermore, the intuitive connection between topology of hyperbolic space and hierarchical structure is clear and it is believable that hierarchical models (like decision trees) may be well poised to dominate in this area.

On the other hand, the paper lacks in convincing evidence that the proposed approach is very effective. The performance gains over other methods are marginal, and the number of repetitions of experiments is very few. The referenced work of Chlenski et al. (2023), which includes another RF model for hyperbolic space is also not included in these experiments, which is a major limitation. Furthermore it is unclear if the hyperparameter tuning has been done in an appropriate manner or not.
On the methodological side in terms of the model structure asopted, it isn't immediately clear that horospheres are the most natural partitioning functions to exploit the "hierarchical" structure of hyperbolic space. If my understanding of this is accurate then something like the intersection of a cone anchored at the origin with an origin centered sphere of radius < 1 seems far more natural. It is also not immediately clear how much novelty there is in the class merging heuristic, nor the weighting term included in the objective to accommodate class imbalance.

---

> ### Author Response · Authors · 2024-04-05
>
> We thank the reviewer for their positive comments regarding the place of our work and its potential, along with their detailed feedback for improving the paper, which we respond to below.
>
> The motivation behind our choice of horospheres is twofold. First, horospheres are inherently easy to work with in hyperbolic space and a natural choice for a first foray into hyperbolic random forests due to their similarities with the Euclidean hyperplane. Second, large-margin classifiers based on horospheres guarantee a globally optimal solution and are the current state-of-the-art for hyperbolic SVMs (Fan et al., 2023), making them a compelling building block.
>
> We disagree the performance gains of our method over the other classifiers are marginal; unlike all baselines, our method never severely underperforms and consistently scores high, especially compared to previous hyperbolic classifiers. For instance, we improve upon the previous state-of-the-art hyperbolic classifier on the nested multi-class WordNet experiment by 35.7 macro-f1. We use a similar number of iterations as used in previous works (e.g., Fan et al., 2023).
>
> # Class merging heuristic novelty
> Our class merging heuristic is new and takes inspiration from hierarchical clustering (Chami et al. 2020) and class merging in oblique random forests (e.g., Katuwal \& Suganthar 2018). We have added a short discussion to the related work.
>
> # Evaluation protocol
> For the established experiments, we use their standard experimental protocol. For all experiments we designed, hyperparameter selection is done on the validation set without any access to the test error; please refer to the general response regarding more details on the hyperparameter selection.
>
> # Concurrent work
> We have added a comparison to the concurrent work of Chlenski et al. in section 6.4; please refer to the general response for a more detailed comment.
>
> # Typos/minor points
> We thank the reviewer for pointing out the typos and minor points and have fixed/incorporated them in the updated version.

---

> > ### Comment · Reviewer_H8RP · 2024-04-10
> > **Thanks, concerns addressed**
> >
> > Thanks to the authors for their extra work and explanations. I have no further concerns

---

### Review · Reviewer_4Yct · 2024-03-14

**Summary Of Contributions:**

The paper presents a method for generalizing random forests from Euclidean to Hyperbolic spaces. The main idea is to find candidate horospheres using a large-margin classifier. They extend the method to multiclass data by combining classes based on the lowest summon ancestor. To further handle imbalanced classes, they introduce a class-balanced large-margin loss. They perform several experiments demonstrating the merits of the new method compared to the existing hyperbolic method or Euclidean tree-based models.

**Audience:**

Yes

**Broader Impact Concerns:**

There are no specific concerns for ethical impications

**Claims And Evidence:**

Yes

**Requested Changes:**

Requested changes and questions

Can you expand on the metrics presented in Table 4? Those are not intuitive

How are the hyperparameters of all baselines tuned?

In the multiclass WorldNet how many splits do you create? One?

On World Net you write: “we split the data into 80% for training and 20% for testing and report the best results from a grid search over three runs in AUPR”
So, what standard deviation is reported?

In general, the evaluation seems different between each dataset, and it isn’t clear how model selection is performed without harming the performance evaluation of the cross-validation procedure [1].


What are the limitations of the method? I don’t see this addressed in the paper.

I would also expect to see a synthetic evaluation to evaluate the performance and limitations of the method in a controlled (hyperbolic) space.


Why is equation 16 broken into two lines? Won’t it fit in one line?

Some basic details on the datasets are missing.

**Strengths And Weaknesses:**

Strengths

The paper is well-written and easy to understand. I found it enjoyable to read. The method proposed in the paper is intuitive and performs well in practice. The authors have taken care to address common issues found in real data, such as multiple classes and imbalanced data.

They have also conducted an ablation study to demonstrate the importance of each component.

Additionally, illustrations have been included in the paper to enhance its readability.


Weaknesses



Complexity or runtime analysis is lacking.

The evaluation is somewhat limited to datasets with expected hyperbolic structure.

I would expect to see a more thorough evaluation using synthetic data with a known hyperbolic structure. The authors can demonstrate the approach's strengths and limitations compared to other baselines. Also, how does the method work on real-world tabular data?


The authors mention Chlenski et al.'s concurrent work on a similar problem but don’t compare it or explain the main differences between the two methods.

---

> ### Author Response · Authors · 2024-04-05
>
> We thank the reviewer for their positive comments regarding the intuitiveness of our approach and their suggestions for improving the paper, which we respond to below.
>
> # Complexity
> We added a section on complexity in section 4.4, where we discuss the runtime of the HoroSplitter and show two practical ways to mitigate it effectively by reducing the number of trees and/or reducing the number of HoroSVM optimization iterations.
>
> # Tabular data
> We have added results on tabular datasets in the supplementary material and the table below, showcasing that HoroRF can compete with the Euclidean methods, despite being designed for hyperbolic data.
>
> |  | an | oms | pt |
> |-----------|-----------|-----------|-----------|
> LinSVM | 100 $\pm0.0$ | 91.4 $\pm1.1$ | 41.2 $\pm0.0$|
> RBFSVM | 100 $\pm0.0$ | 92.8 $\pm0.0$ | 43.0 $\pm0.0$|
> RF | 97.8 $\pm1.6$ | 91.3 $\pm0.2$ | 47.7 $\pm0.8$|
> OblRF | 100 $\pm0.0$ | 92.3 $\pm0.2$ | 42.4 $\pm0.9$|
> HoroRF | 100 $\pm0.0$ | 90.5 $\pm0.4$ | 44.0 $\pm1.1$|
>
> # Synthetic data
> Our visualization in Figure 4 is done on a synthetic tree dataset with a depth of six and a branching factor of four. We have expanded the analysis and show how the hyperbolic splits of HoroRF enable it to reach higher performance than OblRF, especially at low depths, see also below.
>
> | Depth  | 1 | 2 | 3 | 10 | No limit |
> |-----------|-----------|-----------|-----------|-----------|-----------|
> OblRF Euc | 34.5 $\pm$0.1 | **68.8** $\pm$2.5 | 83.3 $\pm$0.0 | **99.3** $\pm$0.0 | 98.9 $\pm$0.0 |
> OblRF Hyp | 33.6 $\pm$0.7 | 65.6 $\pm$1.3 | 83.2 $\pm$0.3 | 99.1 $\pm$0.0 | 98.9 $\pm$0.0 |
> Base HoroRF | 60.8 $\pm$0.0 | 67.8 $\pm$4.9 | 83.6 $\pm$1.6 | 99.1 $\pm$0.0 | **99.1** $\pm$0.3 |
> HoroRF | **63.1** $\pm$3.2 | 67.4 $\pm$5.1 | **85.5** $\pm$1.8 | **99.3** $\pm$0.0 | 98.9 $\pm$0.3 |
>
> # Concurrent work
> We have added a comparison to Chlenski et al.in section 6.4 and a more thorough description of the differences between the methods; please also refer to the general response.
>
> # Hierarchical metric details
> We have expanded on the hierarchical evaluation metrics used in the comparison in section 4.4.
>
> # Evaluation protocol
> Please refer to the general response regarding the evaluation protocol.
>
> # Limitations
> We have added a section on the limitation of the runtime in section 4.4, along with strategies for how to mitigate it in practice.
>
> # Dataset details
> We have expanded section 6.1 in the appendix to include more details for all the datasets used throughout the paper.

---

> > ### Comment · Reviewer_4Yct · 2024-04-15
> > **Response to authors**
> >
> > I thank the authors for the time and effort spent addressing all the comments the reviewers raised.
> > I have read all the new changes by the authors, and my concerns have been addressed.

---

### Review · Reviewer_vu6A · 2024-03-26

**Summary Of Contributions:**

This paper proposes splitting data points based on level-wise horospheres at each node of a tree in a tree ensemble model, with the assumption that data points are embedded into hyperbolic space beforehand. More specifically, the proposed method performs horoSVM, an existing hyperbolic SVM, at each node for splitting. The performance of the resulting ensemble model has been evaluated on real-world datasets embedded into hyperbolic space, comparing it with Euclidean and hyperbolic classifiers.

**Audience:**

Yes

**Broader Impact Concerns:**

I do not have any concerns.

**Claims And Evidence:**

No

**Requested Changes:**

Please address the concerns outlined in the Weaknesses section above.

**Strengths And Weaknesses:**

### Strengths

- Implementing hierarchical splitting based on horospheres within a tree ensemble model represents a novel approach.
- The proposal is carefully designed, amalgamating horoSVM outputs through tree structures. The consideration of class imbalance adds further value.
- Visualizations greatly enhance comprehension of the method.

### Weaknesses

- In this paper, it is consistently assumed that datasets are embedded into hyperbolic space beforehand, and each classifier is applied to the embedded data. However, it is not reasonable to apply non-hyperbolic methods, such as the standard SVM and random forest, to such embedded data. Instead, they should be applied to raw datasets (or with some more appropriate preprocessing). For instance, I cannot understand why axis-aligned splitting and oblique splitting are applied to data points embedded into hyperbolic space in Figure 1. Similarly, in Figure 4, oblique splitting should be visualized in the original data space.
- Regarding the aforementioned issue, I believe Euclidean methods should indeed be conducted on non-hyperbolic datasets as well. Although the authors mention that "the Euclidean baselines are run on the same embeddings as the hyperbolic methods, which we also find to perform better compared to mapping the embeddings back to Euclidean space first", it is crucial to present the full set of experimental results for a thorough evaluation.
- I feel that employing only 100 trees for ensemble models might be somewhat limited. It would be beneficial to conduct experiments with varying numbers of trees, such as 10, 100, and 1000, to thoroughly explore the impact of the ensemble size on model performance.
- Improving the presentation of the paper can indeed enhance its clarity. Here are some suggestions:
	- Ensure consistency in notation: Replace instances of $x$ with $\mathbf{x}$, especially where $\mathbf{x}$ denotes vectors or matrices. Consistency in notation helps maintain clarity and reduces confusion for readers.
	- Equation (7) should be written in one line to avoid ambiguity.

---

> ### Author Response · Authors · 2024-04-05
>
> We thank the reviewer for their positive comments regarding the novelty and careful design of our method and respond to the remaining points below.
>
> # Euclidean methods on Euclidean embeddings
> For all the main experiments, the raw/original data space in this case is, in fact, hyperbolic. It is standard procedure to apply the Euclidean methods to these embeddings as a baseline (e.g., Cho et al., 2019; Fan et al., 2023) due to the success of hyperbolic embeddings in representing hierarchical spaces. Nonetheless, we show in section 6.4 that, for all benchmarks, running the Euclidean methods on the hyperbolic embeddings is often better than using them on the embeddings mapped to Euclidean space. We show the comparison on binary WordNet also in the table below. Furthermore, we have added a visualization of the splits of OblRF on the data mapped to Euclidean space in Figure 4, with conclusions very similar to using OblRF on the raw hyperbolic data.
>
> |  | Animal | Group | Worker | Mammal | Tree | Solid | Occupation | Rodent |
> |-----------|-----------|-----------|-----------|-----------|-----------|-----------|-----------|-----------|
> **Euclidean embeddings**| |
> LinSVM | 2.5  $\pm0.0$   | 5.6  $\pm0.1$   | 0.7  $\pm0.0$   | 0.7  $\pm0.0$   | 0.6  $\pm0.0$   | 0.8  $\pm0.0$   | 0.2  $\pm0.0$   | 0.1  $\pm0.0$ |
> RBFSVM | 2.5  $\pm0.0$   | 5.4  $\pm0.0$   | 0.7  $\pm0.0$   | 0.7  $\pm0.0$   | 0.6  $\pm0.0$   | 0.8  $\pm0.0$   | 0.2  $\pm0.0$   | 0.1  $\pm0.0$ |
> RF | 98.5  $\pm$0.2   | 96.4  $\pm$0.1   | 72.5  $\pm$1.0   | 98.4  $\pm$1.0   | **77.2**  $\pm$2.3   | 91.2  $\pm$0.8   | 63.4  $\pm$0.6   | 28.6  $\pm$6.5|
> OblRF | 98.1  $\pm$0.1   | 96.2  $\pm$0.2   | 71.4  $\pm$0.8   | 98.8  $\pm$0.7   | 77.0  $\pm$1.6   | 91.4  $\pm$0.7   | 62.9  $\pm$0.3   | 29.3  $\pm$7.4 |
> **Hyperbolic embeddings**| |
> LinSVM | 2.5  $\pm0.0$   | 6.0  $\pm0.6$   | 0.7  $\pm0.0$   | 0.7  $\pm0.0$   | 0.6  $\pm0.0$   | 0.8  $\pm0.0$   | 0.2  $\pm0.0$   | 0.1  $\pm0.0$ |
> RBFSVM | 2.5  $\pm0.0$   | 6.0  $\pm0.6$   | 0.7  $\pm0.0$   | 0.7  $\pm0.0$   | 0.6  $\pm0.0$   | 0.8  $\pm0.0$   | 0.2  $\pm0.0$   | 0.1  $\pm0.0$ |
> RF | **98.6**  $\pm$0.2   |  96.6  $\pm$0.4   | 70.0  $\pm3.1$   | **99.3**  $\pm$0.3   | 75.8  $\pm2.0$   | 92.5  $\pm$ 1.2   | 59.8  $\pm1.1$   | 34.5  $\pm3.0$|
> OblRF | 98.4  $\pm$0.1   | **96.7**   $\pm$0.3   | **73.3**  $\pm$1.3   | 98.8  $\pm$0.3   | 76.2  $\pm$3.2   | **92.9**  $\pm$1.7   | **65.0**  $\pm$2.4   | **38.2**  $\pm$0.9|
>
> # Number of trees
> We have added Figure 5a to the manuscript to show the effect of the number of trees on the performance and discuss in section 4.4 that 100 trees are more than sufficient to reach the maximal performance. In fact, fewer trees would reduce the computational cost while keeping performance more or less equal.

---

> > ### Comment · Reviewer_vu6A · 2024-04-11
> >
> > Thank you for addressing the issues. I understand the motivation and evaluation of the paper, and I have no further concerns.

---

### Author Response · Authors · 2024-04-05

We appreciate the reviewers pointing out that our work "represents a novel approach" and is "carefully designed" (vu6A), "intuitive" and "enjoyable to read" (4Yct), and "(placed) in a relevant context" (H8RP). We will address common points below and expand on specific remarks in the individual responses. We have updated the manuscript with all changes highlighted in blue.

# Comparison with HyperRF
We have added a comparison to HyperRF in section 6.4, where we find that the methods perform comparably on the different benchmarks; shown in the table below. We want to stress that HyperRF (Chlenski et al., 2023) is a concurrent work to this submission, with their preprint appearing after ours.

|  | Football | Polblogs | Worker | Tree | Occupation | Rodent | Same level | Nested | Mean|
|-----------|-----------|-----------|-----------|-----------|-----------|-----------|-----------|-----------|-----------|
| HyperRF | 33.0 $\pm$2.3 | 84.4 $\pm$1.8 | 71.5 $\pm$ 0.9 |  **77.5** $\pm$ 2.1 | 53.9 $\pm$ 4.4 | **40.3** $\pm$ 1.6 | **91.7** $\pm$1.0 | **93.8**  $\pm$0.8 | 68.3 |
HoroRF | **38.3** $\pm$1.8 | **86.1** $\pm$1.0 | **73.4** $\pm$ 1.7 |  76.6 $\pm$ 3.8 | **65.6** $\pm$ 1.5 | 39.0 $\pm$ 2.4 | 91.3 $\pm$0.3 | 93.3 $\pm$1.1 | **70.4**|

All results in our paper are obtained by fitting the classifiers to the same embeddings. This approach allows us to fairly compare algorithms, rather than embeddings, for hyperbolic classification. In contrast, HyperRF uses a different model of hyperbolic space, and we have to transform the embeddings. As a result, we are no longer directly comparing classifiers. Overall, we find that HoroRF performs best compared to all classifiers used on the Poincar\'e embeddings with curvature one.

Note also that conclusions drawn from our experimental results differ from Chlenski et al., 2023 as they (1) limit the tree depth in their evaluations to 3 and (2) use the default settings of our open-source implementation, which did not use class balancing at the time of their work.


# Evaluation protocol
For the established experiments (WordNet subtree classification and network node classification), we use the standard experimental setting used in the literature (e.g., Chlenski et al., 2023; Cho et al., 2019; Fan et al., 2023) and report the cross-validation results. For the experiments we designed (multi-class WordNet, hierarchical classification, tabular datasets, synthetic experiments), we split the data into train, validation, and test sets. We optimize the hyperparameters with a grid search on the validation set to select the optimal configuration and report the performance on the test set. As the classifiers are not deterministic, we report the standard deviation over multiple different seeds with the same data split.

# Typos and consistency
We appreciate the feedback of the reviewers regarding the typos and notational consistency. We updated the manuscript accordingly and conducted a thorough proofreading to rectify the issues.

---

### Decision · Action_Editor_voSX · 2024-05-14

**Recommendation:** Accept as is

**Comment:**

All three reviewers proposed Accept/Leaning Accept. Reviewers highlighted that the horospheres may not be the ideal partitioning functions, however the authors have adequately justified their decision.

**Audience:**

Yes.

**Claims And Evidence:**

The manuscript presents a method for generalizing random forests from Euclidean to Hyperbolic spaces. The main idea is to find candidate horospheres using a large-margin classifier. The proposed method can handle multiclass data and data with imbalanced classes. Evidence is provided through experiments comparing the proposed method to existing hyperbolic classifiers and Euclidean counterparts of the hyperbolic classifiers. During the rebuttal period, additional evaluations are performed on tabular data, and further comparisons with HyperRF (Chlenski et al., 2023), a concurrent work, are provided.